# Energy consumption forecasting for oil and coal in China based on hybrid deep learning

Jiao He[1], Yuhang Li[2], Xiaochuan Xu[3], Di Wu[2]*

1 School of International Business and Management, Sichuan International Studies University, Chongqing, China, 2 College of Computer and Information Science, Southwest University, Chongqing, China, 3 State Grid Chongqing Electric Power Company, Chongqing, China

* wudi.cigit@gmail.com

**Data Availability Statement:** The data used in this study is available at the link: https://github.com/wudi1989/Energy_datasets/tree/main.

**Funding:** Science and Technology Foundation of State Grid Corporation of China under grant 1400-

## Abstract

The consumption forecasting of oil and coal can help governments optimize and adjust energy strategies to ensure energy security in China. However, such forecasting is extremely challenging because it is influenced by many complex and uncertain factors. To fill this gap, we propose a hybrid deep learning approach for consumption forecasting of oil and coal in China. It consists of three parts, i.e., feature engineering, model building, and model integration. First, feature engineering is to distinguish the different correlations between targeted indicators and various features. Second, model building is to build five typical deep learning models with different characteristics to forecast targeted indicators. Third, model integration is to ensemble the built five models with a tailored, self-adaptive weighting strategy. As such, our approach enjoys all the merits of the five deep learning models (they have different learning structures and temporal constraints to diversify them for ensembling), making it able to comprehensively capture all the characteristics of different indicators to achieve accurate forecasting. To evaluate the proposed approach, we collected the real 880 pieces of data with 39 factors regarding the energy consumption of China ranging from 1999 to 2021. By conducting extensive experiments on the collected datasets, we have identified the optimal features for four targeted indicators (i.e., import of oil, production of oil, import of coal, and production of coal), respectively. Besides, we have demonstrated that our approach is significantly more accurate than the state-of-the-art forecasting competitors.

## 1. Introduction

China is the world's largest importer and one of the largest consumers of oil and coal. Consumption forecasting of oil and coal is crucial in China as it not only provides a clear understanding of the future energy landscape but also helps the government to optimize and adjust strategies, thereby ensuring energy security [1]. For example, with China's economy's growth, oil and coal consumption has been gradually increasing [1, 2]. However, China is facing the challenge that its reserves of oil and coal are not abundant. Since becoming a net oil importer in 1993, China's external dependence on oil has exceeded 65% [3]. China's reliance on energy

202357341A-1-1-ZN (Identification of Energy Security Risks and Strategic Path Optimization Technology Research under Global Coal-Oil-Gas-Electricity Coupling in China). The funders had no role in study design, data collection and analysis, decision to publish, or preparation of the manuscript.

**Competing interests:** The authors have declared that no competing interests exist.

has steadily risen, making it the world's largest energy-importing country. Hence, forecasting the energy consumption of oil and coal can help China develop its energy-importing strategies to ensure energy security [4].

In principle, the forecasting of consumption of oil and coal is a time series forecasting problem. To date, numerous time series forecasting methods have been proposed [5], including statistical analysis-based [6], machine learning-based [6], and deep learning-based ones [7]. First, autoregressive integrated moving average (ARIMA), exponential smoothing, and grey forecasting models have emerged as notable statistical analysis-based methods for time series forecasting. In terms of ARIMA, it has shown promising results in forecasting future electricity consumption [8, 9]. For example, ARIMA was combined with bootstrap aggregation and exponential smoothing to achieve remarkable performance in mid to long-term electricity consumption forecasting [10]. In terms of exponential smoothing, the double exponential smoothing model was employed to forecast the demands for coal, oil, natural gas, and primary electricity [11]. In terms of grey forecasting models, a grey forecasting model with simulated annealing exhibited higher accuracy than traditional grey models in forecasting coal consumption [12]. In addition, the grey Lotka-Volterra model was built based on the competition and cooperation mechanism to forecast energy consumption [13]. However, these statistical analysis-based methods have their own limits. For example, the exponential smoothing method cannot detect inflection points in data and the grey forecasting model is sensitive to outliers. Furthermore, ARIMA, exponential smoothing, and grey forecasting methods heavily rely on historical data. If the historical data exhibits substantial variability, they may not be suitable for forecasting long-term time series.

Second, with the rapid development of machine learning, it has been employed to forecast time series in various industrial applications [6]. Among various machine learning approaches, Prophet, support vector machines (SVM), and artificial neural networks (ANNs), have gained significant attention because of their feature mappings learning capability between input and output data [14]. For example, the Prophet model performs better in forecasting India's monthly total energy demand and peak energy demand than traditional models [15]. The advanced SVM achieves significantly higher accuracy than traditional models in forecasting solar and wind energy for most regions [16]. ANNs exhibit excellent performance in both real-time and short-term solar energy forecasting [17]. However, these machine learning-based methods are shallow models. They fail to effectively explore the deep potential correlations between various features that are closely related to the forecasting targets.

Finally, by increasing the number of hidden layers in the neural network, deep learning methods excel at handling strong nonlinear deep characteristics with remarkable performance [18]. Recurrent neural networks (RNN) and long short-term memory (LSTM) are two commonly adopted deep learning algorithms for time series forecasting [15]. RNN was first introduced in 1990 to retain temporal information by incorporating a recurrent layer to decide whether to retain information from previous time steps [12]. It outperforms the algorithms employed by the Estonian energy regulatory authority in forecasting wind power generation [19]. However, RNN struggles to maintain long-term dependencies due to the exploding/vanishing gradient problem [20]. To address this issue, LSTM was proposed by designing the gate structure [21]. Besides, it preserves temporal correlation through the utilization of memory cells [22]. Hence, it has good generalizability in forecasting energy-related time series [14].

However, the consumption demands of oil and coal are influenced by many factors [1, 23, 24], such as economy, population, climate, international situation, natural environment, etc. As a result, it is very challenging to investigate which features are most crucial for accurate forecasting. Besides, different energy consumption indicators (e.g., import of oil, production of oil, import of coal, and production of coal) have different inherent characteristics, and a

single deep learning model can not comprehensively capture all the inherent characteristics of different indicators, resulting in its limited forecasting robustness.

Ensemble learning is an effective approach to enhance the forecasting performance of a single model [25]. Instead of relying on one single model, ensemble learning combines a collection of diverse models to create a more robust and accurate forecasting. Recently, ensemble learning has been applied to energy areas. For example, ensemble learning has shown quite promising results in addressing the issue of forecasting building energy consumption [26]. A hybrid forecasting model based on a selective ensemble showed a significant impact in addressing the forecasting issues related to energy consumption in China [27]. Furthermore, ensemble learning exhibited a significant accuracy improvement in predicting the electricity consumption of office buildings [28, 29]. Therefore, these studies demonstrated that ensemble learning models can deal with the heterogeneity among different forecasting issues. However, the previous studies of ensemble learning still have some limitations in forecasting the energy consumption of oil and coal in China. First, the consumption of oil and coal is influenced by many complex and uncertain factors, previous studies did not distinguish the correlations between targeted indicators and various factors. Besides, previous studies did not design a weighting strategy for controlling the ensembling effects, which is unsuitable for complex and uncertain scenarios in forecasting.

Motivated by this, a hybrid deep learning approach is proposed for accurately forecasting the consumption of oil and coal in China. The proposed approach consists of main three parts, i.e., feature engineering, model building, and model integration. First, feature engineering is to adopt correlation analysis to distinguish the different correlations between targeted indicators and various features. Second, model building is to build five typical deep learning models with different characteristics to forecast targeted indicators. Third, model integration is to ensemble the built five deep learning models with a tailored, self-adaptive weighting strategy. As such, the proposed approach enjoys all the merits of the five deep learning models, making it able to comprehensively capture all the characteristics of different factors to achieve accurate energy consumption forecasting. To evaluate the proposed approach, we collected the real 880 pieces of data with 39 factors ranging from 1999 to 2021. Four factors of import of oil, production of oil, import of coal, and production of coal were used as the targeted forecasting indicators because they are closely tied to the energy consumption of China. The remaining 35 factors (e.g., natural gas production, total construction industry output, corn yield, etc.) were used as the features. By conducting extensive experiments, we demonstrated that: a) the significant features of the four targeted indicators were identified respectively, and b) the proposed approach significantly outperforms both state-of-the-art statistical and deep learning comparison models in forecasting the four targeted indicators.

## 2. Methodology

### 2.1 Design philosophy

Fig 1 illustrates the general process of our proposed approach with five steps. The specific steps are outlined as follows:

**Step 1: Input Data.**   Collecting as much as possible data that may be related to energy consumption from economy, culture, society, etc. Split these data into targeted forecasting indicators and relevant features. In this paper, we have collected 39 factors, where four factors (i.e., import of oil, production of oil, import of coal, and production of coal) are used as the targeted forecasting indicators and the remaining are used as the features.

**Step 2: Feature Engineering.**   Conducting correlation analysis based on Pearson correlation coefficient, Spearman correlation coefficient, and Kendall correlation coefficient. The

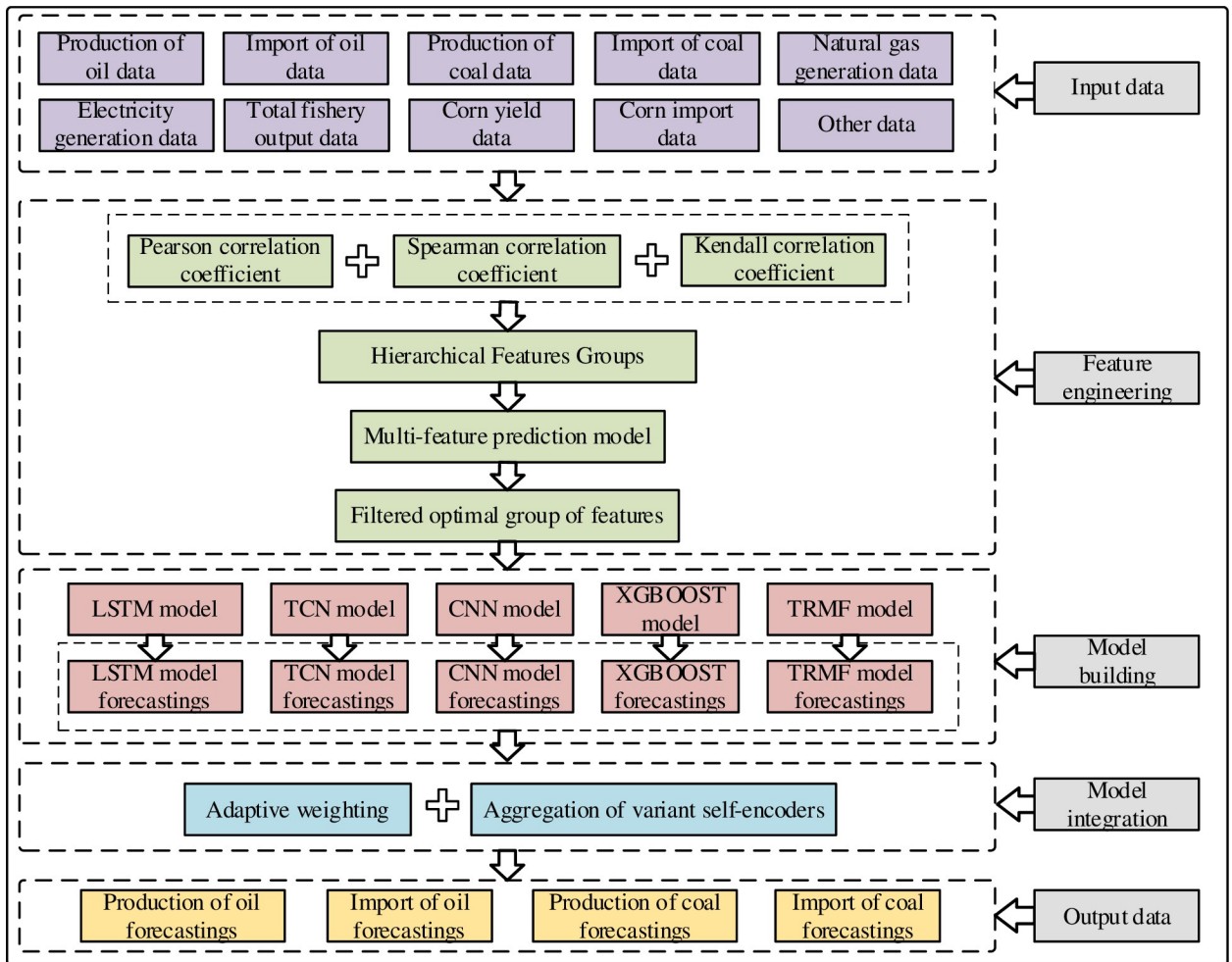

**Fig 1. The flowchart of our proposed approach has five steps, i.e., input data, feature engineering, model building, model integration, and output results.**

final comprehensive correlation is obtained by taking the weighted average of the coefficients derived from these methods. Based on the correlation coefficient values, features are grouped according to their correlation coefficients. Moreover, the grouped features are individually input into a forecasting model to further select the best feature groups based on forecasting errors.

**Step 3: Model Building.** Five deep learning models of LSTM (long short-term memory network) [21], XGBOOST (eXtreme Gradient Boosting) [30], TCN (temporal convolutional network) [31], CNN (convolutional neural networks) [32], and TRMF (temporal regularized matrix factorization) [33] are employed as the forecasting models. The selected best feature groups are used to train these forecasting models to obtain the forecasting results.

**Step 4: Model Integration.** The forecasting results from different forecasting models are ensembled through an adaptive weighting strategy to develop a robust forecasting model. As such, the integrated model can accurately forecast all four targeted indicators.

**Step 5: Output Results.** The forecasting results for the four targeted indicators, i.e., import of oil, production of oil, import of coal, and production of coal, obtained from the integrated model are outputted.

## 2.2 Feature engineering

Correlation refers to the degree of association between two or more variables, serving as a measurement in the fields of statistics and data analysis to assess the interdependence among variables [34]. When variable x changes, another highly correlated variable y will also change accordingly. The correlation coefficient, COR(x, y), ranges from -1 to 1, and the closer the absolute value of COR(x, y) is to 1, the stronger the correlation between variables x and y.

Various statistical methods can be utilized to calculate the correlation. Among them, the Pearson correlation coefficient is commonly employed to quantify the linear relationship between two continuous variables. Additionally, several other correlation coefficients are employed, such as the Spearman correlation coefficient, utilized to gauge the relationship between two ordinal variables, and the Kendall correlation coefficient, used to measure non-linear relationships between two variables. These diverse correlation coefficient methods contribute to understanding relationships between variables of different types, thereby enhancing the capability for data interpretation and analysis.

**2.2.1 Pearson correlation coefficient (PCC).**   The Pearson correlation coefficient is a statistical measurement used to quantify the linear relationship between two random variables represented as real-valued vectors. It holds significant historical importance as the formal method for measuring correlation [14]. This linear correlation coefficient is specifically designed to assess the linear correlation between two normally distributed continuous variables, and its definition is as follows [35]:

$$COR(x, y) = \frac{\sum_1^n (x_i - \bar{x}) \sum_1^n (y_i - \bar{y})}{\sqrt{\sum_1^n (x_i - \bar{x})^2 \sum_1^n (y_i - \bar{y})^2}} \tag{1}$$

where $\bar{x} = \frac{1}{n} \times \sum_{i=1}^n x_i$ denotes the mean of $x$, $\bar{y} = \frac{1}{n} \sum_{i=1}^n y_i$ denotes the mean of $y$, and $n$ is the number of variables in each group.

**2.2.2 Spearman's rank correlation coefficient.**   Spearman's rank correlation coefficient is to evaluate the non-linear correlation between two variables. Instead of using their original observed values, it is computed based on the ranks (orderings) of the variables [36]. The Spearman correlation coefficient COR(x,y) is:

$$COR(x, y) = 1 - \frac{6 \sum_{i=1}^n (x_i - y_i)^2}{n^3 - n} \tag{2}$$

The Spearman correlation coefficient is insensitive to the distribution of data and is applicable to various types of data.

**2.2.3 Kendall correlation coefficient.**   The Kendall correlation coefficient serves as a metric to quantify the strength of the ordinal relationship between two variables. Its computation involves a comprehensive comparison of all pairs of observations within the dataset, aiming to discern their inherent ordinal hierarchy [37]. The Kendall correlation coefficient $\tau$ is as follows:

$$\tau = \frac{\varphi_1 - \varphi_2}{\frac{1}{2} n(n - 1)} \tag{3}$$

where $\varphi_1$ denotes the number of corresponding observations in the data where the ordinal relationships are concordant, meaning the ranks are identical in both variables, $\varphi_2$ denotes the number of corresponding observations in the data where the ordinal relationships are discordant, meaning the ranks are different in the two variables. n represents the sample size.

Distinguishing itself from the Pearson correlation coefficient, which emphasizes linear associations, Kendall proves particularly adept at capturing non-linear relationships. Furthermore,

in handling tied ranks, Kendall exhibits enhanced robustness compared to its counterpart, Spearman.

## 2.3 Forecasting models

A single deep learning model can not comprehensively capture all the inherent characteristics of different indicators. Ensemble learning is an effective way to address this issue by combining several deep learning models. However, ensemble learning requires that the base deep learning models have the characteristics of diversity and accuracy. Diversity: the individual models need to make different kinds of predictions, which ensures that when one model makes a wrong prediction, another can potentially correct it. Accuracy: the individual models should still be reasonably accurate. If each model is weak (too inaccurate), the ensemble won't perform well, even with diversity. Following this principle, we employed the five typical deep learning models of LSTM [21], XGBOOST [30], TCN [31], CNN [32], and TRMF [33] for ensembling. First, they have different learning structures and forecasting mechanisms to make them diverse. Second, they have been demonstrated to be accurate and effective in forecasting energy-related time-series. Hence, the ensembled model enjoys all the merits of the five deep learning models, making it able to comprehensively capture all the characteristics of different indicators to achieve accurate forecasting. Next, we introduce the five models.

**2.3.1 Long short-term memory network (LSTM).** LSTM [21] is a specialized variant of the recurrent neural network. When dealing with sequence data, LSTM is capable of effectively capturing and utilizing long-term dependencies. In comparison to traditional RNN models, LSTM has been more successful in addressing issues like gradient vanishing and explosion.

The LSTM model is composed of a sequence of LSTM units, each containing three crucial gate mechanisms: the Input Gate, Forget Gate, and Output Gate. These gate mechanisms are designed to selectively filter and adjust input data, precisely controlling the flow of information and the transmission of memory. These gates are stored in the memory block. Fig 2 shows the structure of a memory block. The equations for LSTM calculations at time 't' are presented below:

$$
\begin{aligned}
\sigma(x) &= \frac{1}{1 + e^{-x}} \\
i_t &= \sigma(W_i X_t + U_i h_{t-1} + b_i) \\
f_t &= \sigma(W_f X_t + U_f h_{t-1} + b_f) \\
o_t &= \sigma(W_o X_t + U_o h_{t-1} + b_o) \\
g_t &= tanh(W_g X_t + U_g h_{t-1} + b_g) \\
C_t &= f_t \times C_{t-1} + i_t \times g_t \\
h_t &= o_t \times \tanh(C_t)
\end{aligned}
\tag{4}
$$

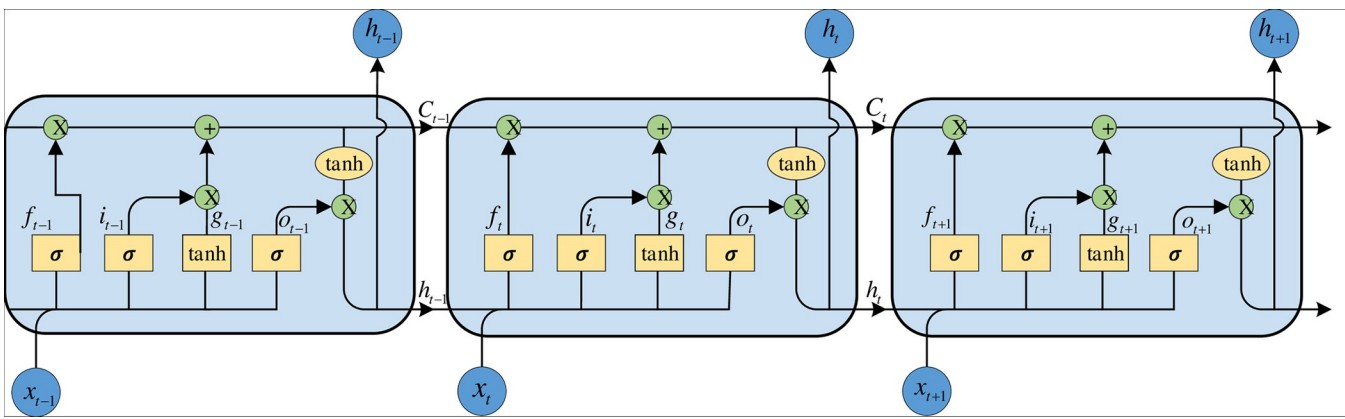

**Fig 2. The architecture of the memory blocks of LSTM.**

where $i_t, f_t, o_t$ represent the input gate, forget gate, and output gate, respectively. $g_t$ serves as an intermediate value during the computation. The $W_i, U_i, W_f, U_f, W_o, U_o, W_g, U_g$ are weight matrices and $b_i, b_f, b_o, b_g$ are bias vectors. $h_t$ and $h_{t-1}$ constitute the outputs at both the current time t and the preceding time t-1, respectively. $X_t$ is the current input. The hyperbolic tangent functions are as follows:

$$tanh(x) = \frac{e^x - e^{-x}}{e^x + e^{-x}}$$
$$sigmoid(x) = \frac{1}{1 + e^{-x}}$$
$$relu(x) = max(0, x)$$

(5)

**2.3.2 eXtreme gradient boosting (XGBOOST).** XGBOOST [30] is a gradient boosting tree algorithm that iteratively trains multiple decision trees. Each tree corrects the residual errors of the previous one, and their outputs are aggregated for predictions. The model's output is the cumulative sum of the outputs from multiple decision trees, with the weight of each tree controlled by a learning rate.

For the forecasting value of the i-th sample, it can be represented as the accumulation of outputs from all trees:

$$\hat{y}_i = \sum_{k=1}^{K} f_k(x_i)$$

(6)

where K is the number of trees and $f_k(x_i)$ is the output of the $k$-th tree for the sample $x_i$. To train the model, it is necessary to define the loss function and regularization term. The objective of XGBoost is to minimize the following loss function:

$$L(\phi) = \sum_{i=1}^{n} l(y_i, \hat{y}_i) + \sum_{k=1}^{K} \Omega(f_k)$$

(7)

where, loss function $l(y_i, \hat{y}_i)$ measures the difference between the true value $y_i$ and the forecasting value $\hat{y}_i$, while the regularization term $\Omega(f_k)$ is used to control the complexity of each tree $f_k$.

**2.3.3 Temporal convolutional network (TCN).** TCN [31] is a deep learning architecture designed for sequence modeling. Unlike traditional Recurrent Neural Networks (RNN), TCN employs Convolutional Neural Networks (CNN) to capture long-range dependencies within sequences.

Given an input sequence $X = (x_1, x_2, ..., x_T)$, TCN generates an output sequence $Y = (y_1, y_2, ..., y_T)$ through a series of convolutional operations. Each convolutional layer applies an activation function and includes residual connections. The output of TCN can be computed as follows:

$$Y_i = H(X_t) + X_t$$

(8)

where $H(X_t)$ represents the output of the convolutional layer, and $X_t$ is the element of the input sequence. $Y_i$ is the output of the network at time $t$.

TCN introduces residual connections, making the network easier to train and helping to prevent issues like gradient vanishing or exploding. It is particularly effective in handling deep networks.

**2.3.4 Convolutional neural networks (CNN).** The role of CNN [32] in time series forecasting is to enhance the model's understanding of patterns and structures within the sequence through convolutional operations and feature learning, thereby improving predictive

performance. In specific time series problems, the combination or nested use of CNN with other models can also yield promising results.

For an input matrix (or feature map) I and a convolutional kernel (or filter) K, the mathematical expression for the convolution operation is as follows:

$$S(i,j) = (I*K)(i,j) = \sum_m \sum_n I(m,n) \cdot K(i-m, j-n) \tag{9}$$

In this context, $S(i,j)$ represents the outcome of the convolution operation. The coordinates $(i,j)$ signify the position within the resulting matrix, while m and n serve as indices for the convolutional kernel. $I(m,n)$ denotes an element within the input matrix, and $K(i-m, j-n)$ signifies the weight associated with the convolutional kernel.

This mathematical expression articulates that each element in the resulting matrix is obtained by performing a weighted summation of the input matrix and the convolutional kernel, adhering to specific computational rules. This convolutional operation adeptly captures local features inherent in the input data, enabling CNNs to discern spatial structures and patterns present in images.

**2.3.5 Temporal regularized matrix factorization (TRMF).** TRMF [33] is a model designed for time series prediction. It captures the underlying structure in sequences by decomposing the time series data matrix into the product of two low-rank matrices. TRMF introduces regularization terms during the decomposition to prevent overfitting and enhance the model's generalization capability. The simple formula for TRMF can be expressed as:

$$Y_i X_{ijt} = \sum_{k=1}^{K} F_i X_{kj(t-1)} + \sum_{l=1}^{L} W_{il} Y_{lt} + \epsilon_{ijt} \tag{10}$$

where, $X_{ijt}$ is the observed value at time t, variable i, in the time series data matrix. $F$ is the time factor matrix, representing the influence of time. $W$ is the space factor matrix, representing the relationships between variables. $Y_{lt}$ is additional information at time t. $\epsilon_{ijt}$ is the error term.

The training process of the model involves finding appropriate factor matrices ($F$, $W$, etc.) through optimization algorithms to minimize the error between the actual observed values and the model predictions. TRMF excels in time series decomposition and prediction, particularly demonstrating advantages in handling missing data and large-scale datasets.

## 2.4 Ensemble modeling

Adaptive aggregation ensemble learning [38] is an effective method for aggregating multiple models. We employ this approach for model integration. The theoretical foundation for adaptive aggregation ensemble learning has been supported. Let $Err^k(t)$ be the forecasting errors of $k$-th model in the five forecasting models at $t$-th training iteration, where $k \in \{1,2,3,4,5\}$ The adaptive weights $\varepsilon^k(t)$ of $k$-th model at $t$-th training iteration can be expressed as follows:

$$Al^k(t) = \sum_{h=1}^{t} Err^k(h), \quad \varepsilon^k(t) = \frac{e^{-\delta Al^k(n)}}{\sum_{t=1}^{4} e^{-\delta Al^k(n)}} \tag{11}$$

where, $t$ is training iteration, $\delta = \sqrt{1/\ln T}$ is the equilibrium factor that governs the aggregation weights of the ensemble during the training process, $T$ is the maximum training iteration, $Al^k(t)$ is the cumulative errors of $k$-th model over $t$ iterations. The process of model ensemble is illustrated in Fig 3.

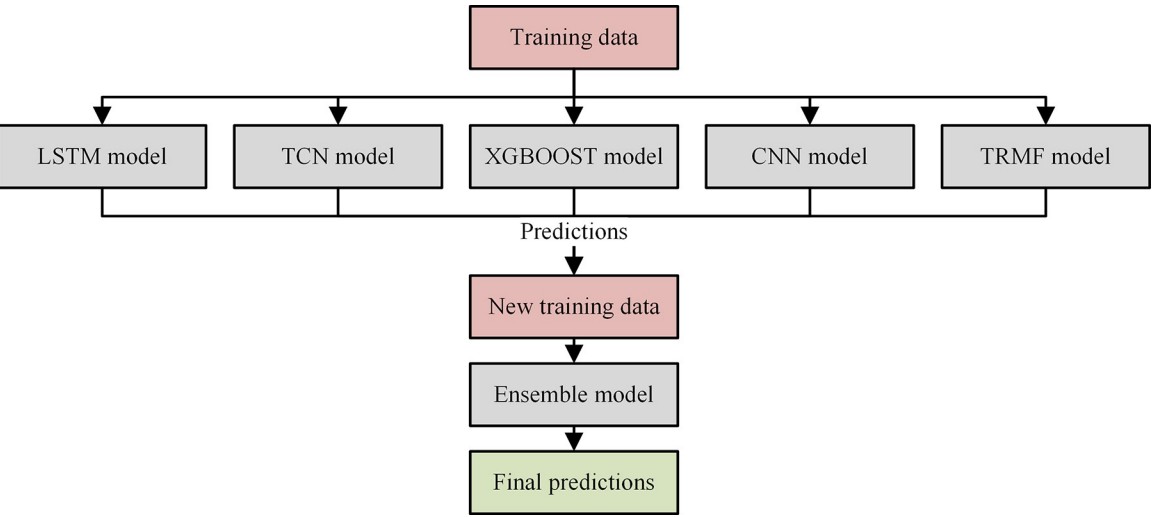

**Fig 3. The process of ensembling five models of our approach.**

## 3. Experiments and results

### 3.1 Data collection

The production data for Chinese oil and coal is sourced from the China Statistical Yearbook (stats.gov.cn) published by the National Bureau of Statistics. The import data for Chinese oil and coal is derived from China's import and export trade data published by the General Administration of Customs(customs.gov.cn). Additionally, we collected more than 50 sets of feature data related to various aspects of the economy, culture, and society from the aforementioned sources. Subsequently, we extracted data for a time period from 1999 to 2021. Finally, we collected four indicators (import of oil, production of oil, import of coal, and production of coal) with total 88 data entries, and 35 features (natural gas production, electricity production, total construction industry output, etc.) with total 770 data entries. More details can be found in the S1 Appendix.

### 3.2 Experimental settings

**3.2.1 Experiment design.** To evaluate the proposed approach, we design four sets of experiments, i.e., feature engineering, comparison between multi-feature trained model and univariate models, comparison between single model and our proposed ensemble approach, and hyperparameter analysis of forecasting models. First is feature engineering, it calculates three correlation coefficients between indicators and features to group the features based on the obtained correlation coefficients. Then, inputting the grouped features into a multi-feature time series forecasting model to identify the optimal features. Due to the limited page space, only LSTM is employed to identify the optimal features in the experiments. Second is comparison between multi-feature trained model and univariate models, it compares the model trained on the identified optimal features with several univariate models to validate that multi-feature data yields better forecasting accuracy than univariate data. Third is comparison between single model and our proposed ensemble approach, it compares the single model with out proposed ensemble model to demonstrate that our approach can improve the forecasting accuracy of each single model on all four indicators. Fourth is hyperparameter analysis of forecasting models, it conducts hyperparameter sensitivity analysis of LSTM to show that a

single model is sensitive to hyperparameters and then monitores the ensemble weights and convergence curves of ensemble model to show that our proposed approach is adaptive to achieve better forecasting performance.

**3.2.2 Evaluation metric.** In the experiments, the training sets were constructed by using the data from 1999 to 2016, and the remaining five years data (i.e., 2017, 2018, 2019, 2020, and 2021) were set as testing sets. The model hyperparameters were tuned by grid search. We repeated each experiment five times to obtain the average forecasting accuracy. Root mean square error (RMSE) and mean absolute percentage error (MAPE) are commonly adopted to assess forecasting accuracy in various related time series forecasting scenarios [9]. The computing formulas of RMSE and MAPE are given as follows:

$$RMSE = \sqrt{\frac{1}{N}\sum_{i=1}^{N}(Y_i - \hat{Y}_i)^2}, \quad MAPE = \frac{100}{N}\sum_{i=1}^{N}|\frac{Y_i - \hat{Y}_i}{Y_i}| \tag{12}$$

where $N$ represents the count of data samples in the testing set, $Y_i$ represents the real value of the $i$-th sample, and $\hat{Y}_i$ represents the forecasting value of the $i$-th sample. Lower RMSE and MAPE correspond to a higher forecasting accuracy.

## 3.3 Feature engineering

To identify the optimal features for the four targeted indicators (import of oil, production of oil, import of coal, and production of coal), we initially computed three types of correlation coefficient (i.e., Pearson correlation coefficient, Spearman correlation coefficient, and Kendall correlation coefficient) between the four targeted indicators and the 35 features. Then, the three types of correlation coefficients were averaged to obtain the final averaged correlation coefficients. We grouped the features based on the final averaged correlation coefficients as shown in Tables 1–4 (the detailed results of the correlation coefficients can be found in Tables B1-B4 in the S1 Appendix).

**Table 1. The results of grouped features with different averaged correlation coefficients for the indicator of production of oil.** There are six groups divided by the correlation coefficient range, i.e., the correlation coefficient ranges of 0.90–1.00, 0.80–0.90, 0.70–0.80, 0.60–0.70, 0.50–0.60, and 0.00–0.50, respectively.

| Correlation coefficient range | Features |
|---|---|
| 0.90–1.00 | None |
| 0.80–0.90 | Production of coal, number of residents under minimum living guarantee, labor force. |
| 0.70–0.80 | Import of coal, electricity generation, total fishery output, total livestock output, total profits of large-scale industrial enterprises, actual utilized foreign investment amount, corn yield, soybean import volume, number of graduates from regular higher education institutions, number of admissions to regular higher education institutions, number of book publications, number of participants in maternity insurance, number of participants in work-related injury insurance, employment rate, engel coefficient of residents. |
| 0.60–0.70 | Import of oil, natural gas generation, total construction industry output, total agricultural output, total forestry output, wheat yield, rice yield, number of participants in unemployment insurance, number of units in social service institutions, index of resident consumption level, research and development (R&D) expenditure. |
| 0.50–0.60 | corn import volume, rice import volume, research and development (R&D) expenditure growth rate. |
| 0.00–0.50 | Number of newly established foreign-invested enterprises, soybean generation, barley import volume, wheat import volume, natural population growth rate of permanent residents, birth rate of permanent residents. |

**Table 2. The results of grouped features with different averaged correlation coefficients for the indicator of import of oil.** There are seven groups divided by the correlation coefficient range, i.e., the correlation coefficient ranges of 0.99–1.00, 0.90–0.99, 0.80–0.90, 0.70–0.80, 0.60–0.70, 0.50–0.60, and 0.00–0.50.

| Correlation coefficient range | Features |
|---|---|
| 0.99–1.00 | Natural gas generation, electricity generation, total construction industry output, total forestry output, number of participants in unemployment insurance, index of resident consumption level. |
| 0.90–0.99 | total fishery output, total livestock output, total agricultural output, total profits of large-scale industrial enterprises, actual utilized foreign investment amount, corn yield, soybean import volume, number of graduates from regular higher education institutions, number of admissions to regular higher education institutions, number of book publications, number of participants in work-related injury insurance, number of units in social service institutions, employment rate, number of participants in maternity insurance, engel coefficient of residents. |
| 0.80–0.90 | Import of coal, production of coal, wheat yield, rice yield, rice import volume. |
| 0.70–0.80 | corn import volume, labor force. |
| 0.60–0.70 | Production of oil, barley import volume, wheat import volume, R&D expenditure growth rate. |
| 0.50–0.60 | natural population growth rate of permanent residents, birth rate of permanent residents. |
| 0.00–0.50 | Number of newly established foreign-invested enterprises, soybean generation, number of residents under minimum living guarantee, research and development (R&D) expenditure. |

To identify the optimal features for forecasting, we sequentially added the grouped features into LSTM for forecasting in descending order with different correlation coefficients, as shown in Tables 5–8. By finding the highest forecasting accuracy, the corresponding optimal features can be identified. From Tables 5–8, we obtained the following four aspects of observations:

- Incorporating features with high correlation coefficients can enhance the model's forecasting accuracy in general. For example, the models with added features exhibit higher forecasting accuracy than the "Single feature" group that has no feature. Another example, Table 8 shows that the model's forecasting accuracy is improved by sequentially adding features until the last group (0.00–1.00).

**Table 3. The results of grouped features with different averaged correlation coefficients for the indicator of production of coal.** There are six groups divided by the correlation coefficient range, i.e., the correlation coefficient ranges of 0.90–1.00, 0.80–0.90, 0.70–0.80, 0.60–0.70, 0.50–0.60, and 0.00–0.50.

| Correlation coefficient range | Features |
|---|---|
| 0.90–1.00 | Import of coal. |
| 0.80–0.90 | Import of oil, natural gas generation, Production of oil, electricity generation, total construction industry output, total fishery output, total livestock output, total profits of large-scale industrial enterprises, actual utilized foreign investment amount, corn yield, soybean import volume, number of graduates from regular higher education institutions, number of admissions to regular higher education institutions, number of book publications, number of participants in maternity insurance, number of participants in unemployment insurance, number of participants in work-related injury insurance, employment rate, labor force, index of resident consumption level, Engel coefficient of residents, total agricultural output, total forestry output, research and development (R&D) expenditure, number of units in social service institutions. |
| 0.70–0.80 | wheat yield, rice yield, corn import volume. |
| 0.60–0.70 | Rice import volume, number of residents under minimum living guarantee. |
| 0.50–0.60 | Barley import volume, wheat import volume, Research and development (R&D) expenditure growth rate, natural population growth rate of permanent residents. |
| 0.00–0.50 | Number of newly established foreign-invested enterprises, soybean generation, birth rate of permanent residents. |

**Table 4. The results of grouped features with different averaged correlation coefficients for the indicator of import of coal.** There are six groups divided by the correlation coefficient range, i.e., the correlation coefficient ranges of 0.90–1.00, 0.80–0.90, 0.70–0.80, 0.60–0.70, 0.50–0.60, and 0.00–0.50.

| Correlation coefficient range | Features |
|---|---|
| 0.90–1.00 | Production of coal, natural gas generation, electricity generation, total construction industry output, total fishery output, total agricultural output, total forestry output, number of graduates from regular higher education institutions, number of book publications, number of participants in maternity insurance, number of participants in work-related injury insurance, number of units in social service institutions, employment rate, index of resident consumption level, research and development (R&D) expenditure. |
| 0.80–0.90 | Import of oil, total livestock output, total profits of large-scale industrial enterprises, actual utilized foreign investment amount, corn yield, wheat yield, soybean import volume, number of participants in unemployment insurance, labor force, Engel coefficient of residents. |
| 0.70–0.80 | Production of oil, rice yield, rice import volume, corn import volume. number of admissions to regular higher education institutions. |
| 0.60–0.70 | Wheat import volume, research and development (R&D) expenditure growth rate. |
| 0.50–0.60 | Barley import volume, number of residents under minimum living guarantee. |
| 0.00–0.50 | Number of newly established foreign-invested enterprises, soybean generation, natural population growth rate of permanent residents, birth rate of permanent residents. |

- Incorporating features with relatively high correlation coefficients can also lead to a decrease of forecasting accuracy. For example, Table 7 shows that the average MAPE increases from 1.88% to 2.41% after adding features with correlation coefficients of 0.90–0.99. The reason may be that there are some redundant features within the correlation coefficients of 0.90–0.99. In other words, some features within the correlation coefficients of 0.90–0.99 are covered by the features within the correlation coefficients of 0.99–1.00.

- The production and import of oil and coal are related to the production and import of other energy sources, i.e., incorporating other energy-related data can improve forecasting accuracy. Such improvement could be attributed to the Chinese Energy Conservation and Emission Reduction Strategy introduced in 2006 and the Energy Development Strategic Action Plan formulated in 2014, which continuously optimized the Chinese energy structure.

**Table 5. The forecasting results of MAPE and RMSE by sequentially adding the grouped features of Table 1 into LSTM for the indicator of production of oil.** The grouped features with different correlation coefficients were added in descending order. The column of 'Single feature' denotes that only the targeted indicator (without features) was used to model for forecasting. The columns of 'Grouped features with different correlation coefficient ranges' denotes that the features with corresponding correlation coefficients were used to model for forecasting. For example, the column of '0.90–1.00' denotes that the features with correlation coefficients of 0.90–1.00 were used to model.

| Year | Single feature | Grouped features with different correlation coefficient ranges | | | | | | |
|---|---|---|---|---|---|---|---|---|
| | | 0.99–1.00 | 0.90–1.00 | 0.80–1.00* | 0.70–1.00 | 0.60–1.00 | 0.50–1.00 | 0.00–1.00 |
| 2017(MAPE) | 4.91% | 4.91% | 4.91% | **1.50%** | 2.75% | 0.58% | 1.43% | 0.91% |
| 2018(MAPE) | 4.45% | 4.45% | 4.45% | **0.03%** | 3.34% | 3.36% | 1.01% | 2.87% |
| 2019(MAPE) | 5.52% | 5.52% | 5.52% | **0.31%** | 1.44% | 1.77% | 1.34% | 2.17% |
| 2020(MAPE) | 0.44% | 0.44% | 0.44% | **1.53%** | 1.47% | 1.12% | 1.88% | 0.92% |
| 2021(MAPE) | 1.54% | 1.54% | 1.54% | **2.18%** | 0.82% | 3.17% | 2.96% | 1.05% |
| Average MAPE | 3.37% | 3.37% | 3.37% | **1.11%** | 1.96% | 2.00% | 1.73% | 1.58% |
| Average RMSE | 645.87 | 645.87 | 645.87 | **217.04** | 376.31 | 387.0962 | 335.4972 | 304.24 |

* The achieved highest forecasting accuracy by the features with correlation coefficients of 0.80–1.00.

**Table 6. The forecasting results of MAPE and RMSE by sequentially adding the grouped features of Table 2 into LSTM for the indicator of import of oil.** The grouped features with different correlation coefficients were added in descending order. The column of 'Single feature' denotes that only the targeted indicator (without features) was used to model for forecasting. The columns of 'Grouped features with different correlation coefficient ranges' denotes that the features with corresponding correlation coefficients were used to model for forecasting. For example, the column of '0.90–1.00' denotes that the features with correlation coefficients of 0.90–1.00 were used to model.

| Year | Single feature | Grouped features with different correlation coefficient ranges | | | | | | |
|---|---|---|---|---|---|---|---|---|
| | | 0.99–1.00* | 0.90–1.00 | 0.80–1.00 | 0.70–1.00 | 0.60–1.00 | 0.50–1.00 | 0.00–1.00 |
| 2017(MAPE) | 3.05% | **2.14%** | 3.21% | 3.98% | 1.82% | 4.61% | 1.84% | 13.15% |
| 2018(MAPE) | 8.60% | **0.63%** | 1.36% | 1.80% | 6.26% | 5.84% | 5.46% | 3.52% |
| 2019(MAPE) | 8.73% | **0.70%** | 2.45% | 1.09% | 3.15% | 2.09% | 8.64% | 5.61% |
| 2020(MAPE) | 14.87% | **3.30%** | 1.40% | 3.76% | 2.61% | 5.33% | 7.51% | 6.69% |
| 2021(MAPE) | 3.64% | **2.63%** | 3.64% | 8.65% | 10.76% | 10.30% | 7.23% | 4.88% |
| Average MAPE | 7.78% | **1.88%** | 2.41% | 3.86% | 4.92% | 5.63% | 6.13% | 6.77% |
| Average RMSE | 3919.06 | **937.09** | 1167.70 | 1906.30 | 2436.20 | 2771.27 | 3087.65 | 3222.53 |

* The achieved highest forecasting accuracy by the features with correlation coefficients of 0.99–1.00.

- Incorporating data related to staple foods such as rice, barley, and wheat import volumes can also enhance forecasting accuracy. Besides, the population growth rate and birth rate can also further enhance forecasting accuracy. These effects could be attributed to the increasing demands of food due to Chines population growth, thereby increasing the correlations between these features and energy consumption.

In summary, this set of experiments identifies the optimal features for the four targeted indicators (import of oil, production of oil, import of coal, and production of coal). These optimal features were employed to conduct the subsequent three sets of experiments.

## 3.4 Comparison between multi-feature trained model and univariate models

To explore whether the identified optimal features can enhance the forecasting accuracy of energy consumption demand, we compared the multi-feature trained LSTM with seven univariate forecasting models based on statistical analysis, machine learning, and deep learning. The seven univariate forecasting models were built on the data of univariate targeted

**Table 7. The forecasting results of MAPE and RMSE by sequentially adding the grouped features of Table 3 into LSTM for the indicator of production of coal.** The grouped features with different correlation coefficients were added in descending order. The column of 'Single feature' denotes that only the targeted indicator (without features) was used to model for forecasting. The columns of 'Grouped features with different correlation coefficient ranges' denotes that the features with corresponding correlation coefficients were used to model for forecasting. For example, the column of '0.90–1.00' denotes that the features with correlation coefficients of 0.90–1.00 were used to model.

| Year | Single feature | Grouped features with different correlation coefficient ranges | | | | | | |
|---|---|---|---|---|---|---|---|---|
| | | 0.99–1.00 | 0.90–1.00 | 0.80–1.00 | 0.70–1.00 | 0.60–1.00 | 0.50–1.00* | 0.00–1.00 |
| 2017(MAPE) | 0.96% | 0.96% | 0.57% | 7.38% | 6.58% | 4.80% | **5.39%** | 10.92% |
| 2018(MAPE) | 9.68% | 9.68% | 6.79% | 1.98% | 0.14% | 2.84% | **1.05%** | 0.48% |
| 2019(MAPE) | 13.17% | 13.17% | 5.15% | 4.28% | 3.32% | 0.02% | **0.16%** | 8.03% |
| 2020(MAPE) | 0.96% | 0.96% | 2.73% | 5.10% | 1.79% | 1.28% | **0.15%** | 3.34% |
| 2021(MAPE) | 4.95% | 4.95% | 7.58% | 2.36% | 6.04% | 3.50% | **1.09%** | 3.66% |
| Average MAPE | 5.95% | 5.95% | 4.56% | 4.22% | 3.57% | 2.49% | **1.57%** | 5.29% |
| Average RMSE | 2.28 | 2.28 | 1.78 | 1.59 | 1.37 | 0.94 | **0.57** | 1.99 |

* The achieved highest forecasting accuracy by the features with correlation coefficients of 0.50–1.00.

**Table 8. The forecasting results of MAPE and RMSE by sequentially adding the grouped features of Table 4 into LSTM for the indicator of import of coal.** The grouped features with different correlation coefficients were added in descending order. The column of 'Single feature' denotes that only the targeted indicator (without features) was used to model for forecasting. The columns of 'Grouped features with different correlation coefficient ranges' denotes that the features with corresponding correlation coefficients were used to model for forecasting. For example, the column of '0.90–1.00' denotes that the features with correlation coefficients of 0.90–1.00 were used to model.

| Year | Single feature | Grouped features with different correlation coefficient ranges | | | | | | |
|---|---|---|---|---|---|---|---|---|
| | | 0.99–1.00 | 0.90–1.00 | 0.80–1.00* | 0.70–1.00 | 0.60–1.00 | 0.50–1.00 | 0.00–1.00 |
| 2017(MAPE) | 10.92% | 10.92% | 9.52% | **2.16%** | 7.38% | 3.46% | 8.70% | 2.16% |
| 2018(MAPE) | 3.56% | 3.56% | 4.44% | **2.38%** | 9.15% | 7.14% | 3.26% | 5.50% |
| 2019(MAPE) | 7.22% | 7.22% | 4.13% | **1.01%** | 4.06% | 4.91% | 8.88% | 1.54% |
| 2020(MAPE) | 5.00% | 5.00% | 4.02% | **3.06%** | 2.54% | 5.62% | 0.05% | 2.45% |
| 2021(MAPE) | 6.28% | 6.28% | 0.77% | **1.89%** | 1.24% | 4.91% | 1.82% | 4.68% |
| Average MAPE | 6.60% | 6.60% | 4.58% | **2.10%** | 4.87% | 5.21% | 4.54% | 3.26% |
| Average RMSE | 1935.04 | 1935.04 | 1306.74 | **619.35** | 1390.30 | 1541.25 | 1307.8214 | 968.86 |

\* The achieved highest forecasting accuracy by the features with correlation coefficients of 0.80–1.00.

indicators. Table 9 provides their brief introductions. The comparison results are presented in Tables 10–13, where we find that the multi-feature trained LSTM can achieve better forecasting accuracy than seven univariate forecasting models. The reason is that the univariate forecasting model has limited learning ability, it cannot comprehensively capture the complex characteristics of the four targeted indicators. Specifically, compared with the best univariate forecasting models in Tables 10–13, the multi-feature trained LSTM reduces MAPE by 2.20%, 2.19%, 3.12%, and 1.34%, respectively, across the four targeted indicators forecasting. Hence, we conclude that the identified optimal features can significantly improve the forecasting accuracy of energy consumption in China.

## 3.5 Comparison between single model and our proposed ensemble approach

This set of experiments compared our proposed ensemble approach with the five single models of LSTM [30], XGBOOST [30], TCN [31], CNN [32], and TRMF [33]. All involved models

**Table 9. The adopted seven univariate forecasting models for exploring whether the optimal features can enhance the forecasting accuracy of energy consumption.**

| Model | Introduction |
|---|---|
| Arima [39] | It is a time series forecasting model that combines autoregressive (AR) and moving average (MA) components to make data stationary by performing differential calculations. |
| ETS [40] | It is a time series forecasting model that captures error, trend, and seasonality components to provide accurate forecasting for various types of time series data. |
| CNN [41] | It is a deep learning model used for image recognition and processing tasks, utilizing convolutional layers to automatically learn spatial hierarchies of features. |
| TCN [42] | It is a variant of CNN designed for sequential data like time series, using dilated convolutions to capture long-range dependencies effectively. |
| GM (1,1) [43] | It is a grey system model used for forecasting by converting original data into grey data and generating forecasting based on its differential equation. |
| Prophet [44] | It is a forecasting model developed by Facebook that incorporates seasonality, holidays, and trend changes, making it suitable for time series data with various patterns. |
| LSTM [14] | It is a specialized type of recurrent neural network designed for processing and forecasting sequential data, known for its ability to capture long-range dependencies and mitigate vanishing gradient problems. |

**Table 10. The comparison results between the multi-feature trained LSTM and the seven univariate forecasting models for the indicator of production of oil.** The results show that the multi-feature trained LSTM (i.e., LSTM_Multi) outperforms the seven univariate forecasting models.

| Year | Arima | ETS | CNN | TCN | GM (1,1) | Prophet | LSTM | LSTM_Multi |
|------|-------|-----|-----|-----|----------|---------|------|------------|
| 2017(MAPE) | 14.89% | 4.39% | 4.29% | 9.02% | 15.08% | 4.76% | 4.91% | 1.50% |
| 2018(MAPE) | 4.93% | 2.56% | 8.16% | 4.73% | 14.59% | 6.23% | 4.45% | 0.03% |
| 2019(MAPE) | 5.29% | 3.93% | 5.16% | 4.12% | 11.86% | 16.34% | 5.52% | 0.31% |
| 2020(MAPE) | 0.70% | 3.58% | 3.89% | 1.48% | 8.51% | 9.38% | 0.44% | 1.53% |
| 2021(MAPE) | 0.79% | 2.12% | 6.13% | 5.14% | 5.63% | 9.51% | 1.54% | 2.18% |
| Average MAPE | 5.32% | 3.31% | 5.53% | 4.90% | 11.14% | 9.24% | 3.37% | 1.11% |
| Average RMSE | 1018.06 | 638.93 | 1066.30 | 944.12 | 944.12 | 2138.89 | 645.87 | 217.04 |

**Table 11. The comparison results between the multi-feature trained LSTM and the seven univariate forecasting models for the indicator of import of oil.** The results show that the multi-feature trained LSTM (i.e., LSTM_Multi) outperforms the seven univariate forecasting models.

| Year | Arima | ETS | CNN | TCN | GM (1,1) | Prophet | LSTM | LSTM_Multi |
|------|-------|-----|-----|-----|----------|---------|------|------------|
| 2017(MAPE) | 4.36% | 7.95% | 0.14% | 15.12% | 6.59% | 17.55% | 3.05% | 2.14% |
| 2018(MAPE) | 4.16% | 4.81% | 10.32% | 16.10% | 7.88% | 17.86% | 8.60% | 0.63% |
| 2019(MAPE) | 0.33% | 1.97% | 1.65% | 5.16% | 8.58% | 14.11% | 8.73% | 0.70% |
| 2020(MAPE) | 1.15% | 0.26% | 3.21% | 3.91% | 7.41% | 16.15% | 14.87% | 3.30% |
| 2021(MAPE) | 10.36% | 12.93% | 4.69% | 19.22% | 5.67% | 22.92% | 3.64% | 2.63% |
| Average MAPE | 4.07% | 5.59% | 4.00% | 11.90% | 7.22% | 17.72% | 7.78% | 1.88% |
| Average RMSE | 1970.61 | 2666.4 | 1961.76 | 5673.22 | 3533.06 | 8652.56 | 3919.06 | 937.09 |

**Table 12. The comparison results between the multi-feature trained LSTM and the seven univariate forecasting models for the indicator of production of coal.** The results show that the multi-feature trained LSTM (i.e., LSTM_Multi) outperforms the seven univariate forecasting models.

| Year | Arima | ETS | CNN | TCN | GM (1,1) | Prophet | LSTM | LSTM_Multi |
|------|-------|-----|-----|-----|----------|---------|------|------------|
| 2017(MAPE) | 12.74% | 12.74% | 3.21% | 9.14% | 26.72% | 23.48% | 0.96% | 5.39% |
| 2018(MAPE) | 7.64% | 3.66% | 4.71% | 13.08% | 19.69% | 22.58% | 9.68% | 1.05% |
| 2019(MAPE) | 0.67% | 0.62% | 4.48% | 1.04% | 15.26% | 19.40% | 13.17% | 0.16% |
| 2020(MAPE) | 2.39% | 2.41% | 1.42% | 0.58% | 14.55% | 15.20% | 0.96% | 0.15% |
| 2021(MAPE) | 4.15% | 4.04% | 11.75% | 2.90% | 9.22% | 9.67% | 4.95% | 1.09% |
| Average MAPE | 5.52% | 4.69% | 5.11% | 5.35% | 17.09% | 18.07% | 5.95% | 1.57% |
| Average RMSE | 2.04 | 1.74 | 2.00 | 1.98 | 6.41 | 6.90 | 2.280 | 0.57 |

**Table 13. The comparison results between the multi-feature trained LSTM and the seven univariate forecasting models for the indicator of import of coal.** The results show that the multi-feature trained LSTM (i.e., LSTM_Multi) outperforms the seven univariate forecasting models.

| Year | Arima | ETS | CNN | TCN | GM (1,1) | Prophet | LSTM | LSTM_Multi |
|------|-------|-----|-----|-----|----------|---------|------|------------|
| 2017(MAPE) | 4.57% | 5.75% | 5.75% | 6.49% | 18.91% | 11.67% | 10.92% | 2.16% |
| 2018(MAPE) | 0.43% | 3.21% | 3.21% | 10.96% | 19.35% | 13.77% | 3.56% | 2.38% |
| 2019(MAPE) | 5.62% | 6.67% | 6.67% | 8.57% | 15.07% | 11.94% | 7.22% | 1.01% |
| 2020(MAPE) | 0.88% | 1.32% | 1.32% | 5.65% | 17.81% | 14.92% | 5.00% | 3.06% |
| 2021(MAPE) | 5.69% | 6.08% | 7.80% | 8.69% | 13.23% | 9.66% | 6.28% | 1.89% |
| Average MAPE | 3.44% | 4.60% | 4.95% | 8.07% | 16.87% | 12.39% | 6.60% | 2.10% |
| Average RMSE | 1030.96 | 1364.85 | 1476.80 | 2385.47 | 4951.63 | 3652.40 | 1935.04 | 619.35 |

were trained with the identical identified optimal features. Their hyperparameters were tuned by grid search. LSTM model had a three-layer neural network structure with each layer containing 50 neurons, a batch size of 8, and the Adam optimizer with a learning rate of 0.001 for 100 epochs. CNN model had a two-layer network structure with each layer containing 101 neurons, a batch size of 1, and a learning rate of 0.01 with the SGD optimizer for 100 epochs. TCN model had a batch size of 5 and the Adam optimizer with learning rate of 0.001 for 100 epochs. XGBoost model explored various hyperparameters, including the number of trees in the range [50, 100, 150, 200, 250, 300], learning rates of [0.01, 0.1, 0.2], and maximum tree depths of [1, 2, 3, 4, 5, 6]. TRMF model set the number of factors to 4, the time delay list of [1, 2, 3], the noise variance to 2, and the maximum iterations of 100. Table 14 shows the comparison results. To better understand these comparison results, we conducted statistical analyses of the loss/win, the Wilcoxon signed rank test, and the Friedman's test following prior studies [38, 45]. From these results, we find that our proposed ensemble approach significantly outperforms the five single comparison models. In the total 100 comparison cases (for each of the four indicators, 25 cases were compared, giving a total of 100 comparison cases), our approach only performs worse than the LSTM with one case and outperforms all other models with the remaining 99 cases. Our approach achieves excellent MAPE for all the cases consistently below 0.9%, and the minimum MAPE can go as low as 0.0052%. In conclusion, this set of

**Table 14. The comparison results of MAPE between a single model and our proposed ensemble approach for the four targeted indicators.** The results show that our approach significantly outperforms the five single comparison models.

| Indicator | Year | LSTM | TCN | CNN | XGBOOST | TRMF | Our Approach♦ |
|---|---|---|---|---|---|---|---|
| Production of oil | 2017 | 1.50% | 1.29% | 5.25% | 2.50% | 2.39% | 0.26% |
| | 2018 | 0.03% | 0.45% | 0.03% | 2.24% | 1.87% | 0.0052% |
| | 2019 | 0.31% | 0.41% | 2.03% | 2.57% | 2.54% | 0.063% |
| | 2020 | 1.53% | 2.05% | 3.03% | 2.32% | 2.45% | 0.31% |
| | 2021 | 2.18% | 1.97% | 1.72% | 3.83% | 7.54% | 0.35% |
| Import of oil | 2017 | 2.14% | 1.13% | 2.51% | 9.19% | 3.24% | 0.50% |
| | 2018 | 0.63% | 0.93% | 3.78% | 9.16% | 0.82% | 0.16% |
| | 2019 | 0.70% | 1.38% | 2.48% | 8.66% | 1.07% | 0.21% |
| | 2020 | 3.30% | 2.66% | 2.16% | 6.70% | 3.00% | 0.43% |
| | 2021 | 2.63% | 2.39% | 10.54% | 1.58% | 1.99% | 0.40% |
| Production of coal | 2017 | 5.39% | 5.24% | 12.47% | 3.92% | 6.12% | 0.78% |
| | 2018 | 1.05% | 1.02% | 7.02% | 1.53% | 2.12% | 0.32% |
| | 2019 | 0.16% | 0.96% | 0.23% | 8.26% | 1.46% | 0.046% |
| | 2020 | 0.15%• | 0.50% | 1.06% | 1.40% | 10.54% | 0.21% |
| | 2021 | 1.09% | 1.11% | 2.99% | 10.55% | 13.70% | 0.22% |
| Import of coal | 2017 | 2.16% | 1.81% | 1.86% | 22.63% | 2.53% | 0.51% |
| | 2018 | 2.38% | 1.79% | 6.10% | 3.21% | 4.53% | 0.89% |
| | 2019 | 1.01% | 1.47% | 1.10% | 6.63% | 3.35% | 0.22% |
| | 2020 | 3.06% | 3.60% | 5.13% | 2.06% | 5.77% | 0.41% |
| | 2021 | 1.89% | 1.27% | 4.10% | 18.77% | 1.35% | 0.27% |
| Statistical analysis | loss/win• | 1/19 | 0/20 | 0/20 | 0/20 | 0/20 | **1/99** |
| | *p*-value⋆ | 6.50E-05 | 4.78E-05 | 4.78E-05 | 4.78E-05 | 4.78E-05 | / |
| | F-rank* | 2.925 | 3.05 | 4.375 | 4.9 | 4.7 | **1.05** |

•The case of our approach loses the comparison, i.e., the row of '2020' in 'Production of coal' and the column of 'LSTM'. ⋆ Our approach significantly outperforms the comparison model if the *p*-value is lower than 0.05. *A lower value denotes better forecasting accuracy.

♦ The situations of the real values and the corresponding forecasted values with our approach were visualized in Fig B1 in the S1 Appendix.

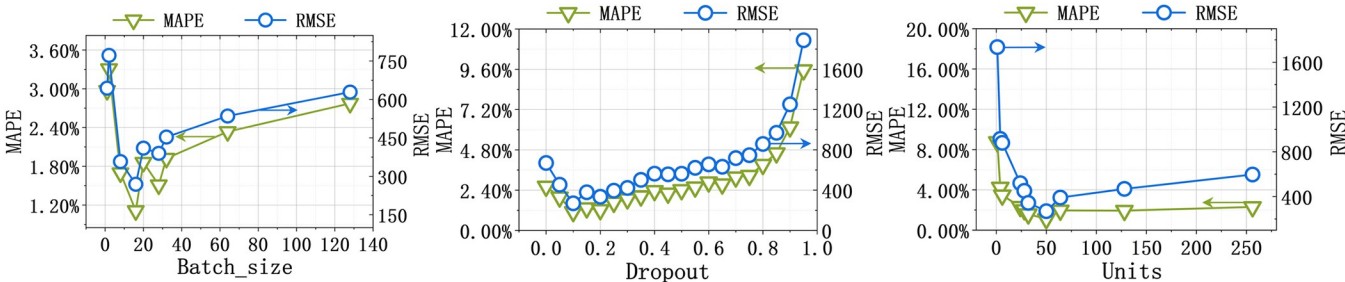

**Fig 4. Hyperparameter analysis of the single LSTM model for the indicator of production of oil shows that the single LSTM model requires manual hyperparameter tunings.**

experiments demonstrates that our proposed ensemble approach can effectively address the limitations of a single model in forecasting heterogeneous targeted indicators of energy consumption.

### 3.6 Hyperparameter analysis of forecasting models

**3.6.1 Hyperparameter analysis of the single LSTM model.** To show that a single model is sensitive to hyperparameters, we conducted the hyperparameter sensitivity analysis of LSTM as a case study. Fig 4 records the results of the production of oil. The complete results on all the datasets are recorded in Fig B2 in the S1 Appendix. From these results, we observed that the single LSTM model is significantly affected by its hyperparameters of batch size, Dropout, and Units. In other words, LSTM requires time-consuming manual hyperparameter tunings for the different forecasting indicators. Therefore, it is challenging for a single model to tune its hyperparameters to accurately forecast all four indicators.

**3.6.2 Hyperparameter analysis of the ensemble model.** To visualize the integration process and performance of the ensemble model, we plot the ensembling weight and convergence curves of our proposed approach. Fig 5 records the adaptive ensembling weights of our approach for 2021. Fig 6 records the convergence curves of all the models for 2021. The results for other years are recorded in Figs B3, B4 in the S1 Appendix. From these figures, we observe that although the convergence patterns of each model vary across different indicators, all the models can consistently converge to better forecasting accuracy with the increasing training epochs. However, our proposed model always achieves the best forecasting accuracy among all the models because it can adaptively adjust its ensembling weights based on the forecasting accuracy. Therefore, the results demonstrate that our approach can adaptively integrate the merits of each single model to obtain an optimal ensemble forecasting performance.

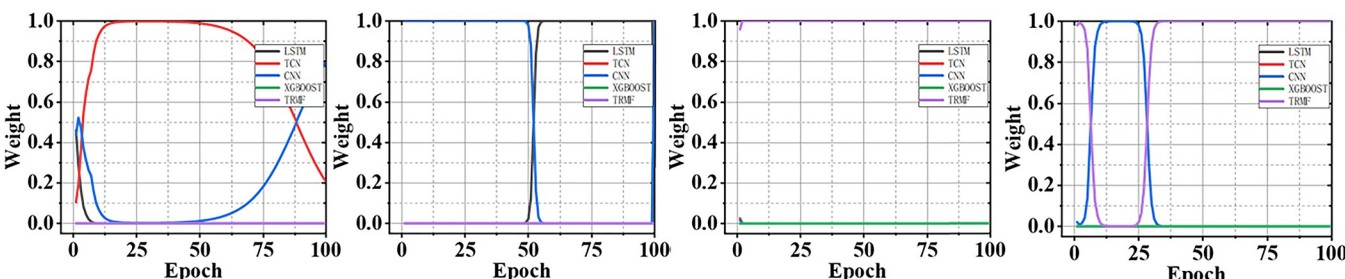

**Fig 5. The adaptive ensembling weights of our approach for 2021.** Four figures are the results of the production of oil, production of coal, import of oil, and import of coal, respectively, from left to right. The results show that our approach can adaptively adjust its ensembling weights.

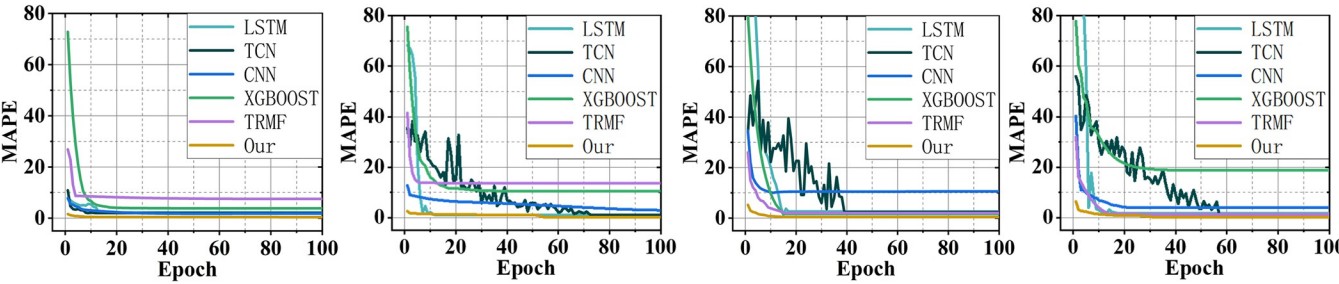

**Fig 6. The convergence curves of all the involved models for 2021.** Four figures are the results of the production of oil, production of coal, import of oil, and import of coal, respectively, from left to right. The results show that all the models can consistently converge to better forecasting accuracy with the increasing training epochs.

## 4. Conclusion

This paper proposes a hybrid deep learning approach for consumption forecasting of oil and coal in China. In the experiments, we collected the real 880 pieces of data with 39 factors regarding the energy consumption of China ranging from 1999 to 2021. The four factors of import of oil, production of oil, import of coal, and production of coal were set as the targeted indicators for representing the energy consumption of China, and the remaining 35 factors were set as features. Based on the experimental results, we have three main kinds of findings. First, feature engineering can not only identify the optimal features for forecasting energy consumption in China but also explore new knowledge. For example, feature engineering discovers that the energy consumption of China is greatly influenced by staple foods, population growth rate, and birth rate. Second, the identified optimal features can significantly improve the forecasting accuracy of energy consumption in China, which indicates that it is critical to find the correct features before modeling. Third, by ensembling five deep learning models, our proposed ensemble approach can effectively address the limitations of a single model in forecasting heterogeneous targeted indicators of energy consumption in China. Note that these ensembling models are required to be diverse with different characteristics [46, 47]. One easy solution is to select the different types of deep learning models, such as the models with different learning principles and structures. Although our approach holds immense potential in energy consumption forecasting, it still has one limitation. The feature engineering of our approach needs to sequentially add the grouped features into a forecasting model to identify the optimal features in advance, which requires some manual adjustments. In the future, we plan to automatically identify the optimal features based on intelligent optimization methods such as differential evolution [45].

## Supporting information

**S1 Appendix.**
(DOCX)

## Author Contributions

**Conceptualization:** Jiao He.

**Data curation:** Yuhang Li, Xiaochuan Xu.

**Formal analysis:** Jiao He, Yuhang Li.

**Funding acquisition:** Di Wu.

**Investigation:** Jiao He, Xiaochuan Xu, Di Wu.

**Methodology:** Jiao He, Yuhang Li, Di Wu.

**Software:** Yuhang Li.

**Supervision:** Di Wu.

**Visualization:** Xiaochuan Xu.

**Writing – original draft:** Jiao He, Yuhang Li.

**Writing – review & editing:** Di Wu.

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
