## [Decision Letter · Decision Letter 0]

1 Aug 2024

PONE-D-24-27898Energy Consumption Forecasting for Oil and Coal in China Based on Hybrid Deep LearningPLOS ONE

Dear Dr. Wu,

Thank you for submitting your manuscript to PLOS ONE. After careful consideration, we feel that it has merit but does not fully meet PLOS ONE’s publication criteria as it currently stands. Therefore, we invite you to submit a revised version of the manuscript that addresses the points raised during the review process.

**Ensure that words such as Novel, new study, first time are removed from the manuscript. Also clarify the meaning of hybrid deep learning methods, all figures should be clear. Authors should also explain the difference between hybrid and stacked ML models. Review for grammatical and spelling errors. **

We look forward to receiving your revised manuscript.

Kind regards,

Jude Okolie, Ph.D.

Academic Editor

PLOS ONE

Journal Requirements:

'Science and Technology Foundation of State Grid Corporation of China under grant 1400-202357341A-1-1-ZN (Identification of Energy Security Risks and Strategic Path Optimization Technology Research under Global Coal-Oil-Gas-Electricity Coupling in China)'

Please state what role the funders took in the study.  If the funders had no role, please state: ''The funders had no role in study design, data collection and analysis, decision to publish, or preparation of the manuscript.'' 

'This work is supported by the Science and Technology Foundation of State Grid Corporation of China under grant 1400-202357341A-1-1-ZN (Identification of Energy Security Risks and Strategic Path Optimization Technology Research under Global Coal-Oil-Gas-Electricity Coupling in China).'

'Science and Technology Foundation of State Grid Corporation of China under grant 1400-202357341A-1-1-ZN (Identification of Energy Security Risks and Strategic Path Optimization Technology Research under Global Coal-Oil-Gas-Electricity Coupling in China)'

Reviewers' comments:

Reviewer's Responses to Questions

**Comments to the Author**

1. Is the manuscript technically sound, and do the data support the conclusions?

Reviewer #1: Partly

Reviewer #2: Yes

2. Has the statistical analysis been performed appropriately and rigorously? 

Reviewer #1: Yes

Reviewer #2: Yes

3. Have the authors made all data underlying the findings in their manuscript fully available?

Reviewer #1: Yes

Reviewer #2: Yes

4. Is the manuscript presented in an intelligible fashion and written in standard English?

Reviewer #1: No

Reviewer #2: No

5. Review Comments to the Author

Reviewer #1: The research presented in this paper is novel and yields significant results, but it requires substantial reshaping to be clear and impactful. In the abstract, the repeated mention of "generation of oil" should be corrected to "generation of coal" to accurately reflect the four targeted indicators. Additionally, more detail on what differentiates the five deep learning models would help readers understand the ensemble's diversity and benefits. Briefly mentioning the types of datasets used, such as the time span and data sources, would also provide clearer context for the experiments and enhance the abstract's informativeness.

The introduction could benefit from a revision of the first sentence for greater clarity, setting a better stage for the research. The data presentation throughout the paper has several issues that need addressing. The constant use of bullet points is distracting; a more narrative style would improve readability and flow. Table 1 is too dense with text, focuses on only one point, and thus is less effective. I do not understnad why that data presented is even in a table. Simplifying and summarizing the information will make it more reader-friendly. Table 2's placement in the paper is confusing and should be reconsidered for better logical flow. Moreover, tables 2 through 5 should be grouped together if they are related, and adding lines to these tables would improve readability. Each table should be accompanied by a general statement explaining its content and significance.

Figures 2 and 5 are blurry and hard to read, necessitating clearer visuals to effectively communicate the data. There is also insufficient explanation for each figure and table; more detailed descriptions rather than the current one-sentence explanations are needed. In Section 3.2.1, the bullet points are unnecessary and should be revised into a more cohesive narrative. Section 3.4 needs to be reworked entirely; the results discussed in the tables should be expanded upon, integrating the discussion into the overall implementation and implications of the work.

The summary requires several improvements. The repeated mention of "generation of oil" should be corrected to accurately reflect the four indicators: import of oil, generation of oil, import of coal, and generation of coal. Additionally, providing a brief explanation or example of what feature engineering involves would be beneficial, as not all readers may be familiar with the term. More detail on the specific deep learning models used in the ensemble would also give a fuller picture of the methodology and its advantages.

By addressing these points, the paper can better communicate its novel approach and significant findings, making it a clearer and more impactful contribution to the field of energy consumption forecasting. I cannot emphasize enough the importance of having more explanation of these tables and figures.

Reviewer #2: Review of Energy Consumption Forecasting for Oil and Coal in China Based on Hybrid Deep Learning

The authors have clearly demonstrated that the use of ensemble models is effective in addressing the limitations of single models in time series forecasting of heterogeneous indicators, including the superiority of using multi-feature data over univariate data in enhancing prediction accuracy. I recommend the acceptance of the paper for publication, subject to some revisions below. Overall, the paper needs some improvement/polishing of the grammar, especially the introduction. The authors should revise the grammar of the manuscript for better readability (I have highlighted JUST A FEW (as “Grammar” below – there is more.)

Introduction

• The second paragraph in this section should be re-written for better flow of thoughts. I find it hard to connect all the different statistical analysis methods and how one relates and/or build on the other. Each sentence in this section sounds disjointed from the preceding and/or proceeding sentence. Consider using words like “nevertheless”, “also”, “however”, “moreover”, “in addition” to connect the various statistical methods, and how they relate to each other.

• Paragraph 6. Rather than simply cite the other previous authors who have implemented ensemble modelling to energy related forecasting, the authors should elaborate more on these previous works (approach, outcomes, shortcoming). This will help build a case for the uniqueness of this article vs previous studies. For example, what makes your approach unique compared to other authors who have implemented ensemble modelling for energy related forecasting?

• Grammar: “Their consumption forecasting is crucial for China because it can not only provide a clear understanding of the future energy situation, but also help governments optimize and adjust its energy strategies to ensure energy security”, can be re-written as “Consumption forecasting is crucial in China as it not only provide a clear understanding of the future energy landscape but also helps the government to optimize and adjust strategies, thereby ensuring energy security”

• Grammar: Ensemble model combines a team of diverse models…Ensemble model combines a collection of diverse models

Section 2.3

• The authors have done an excellent job in describing the five forecasting models. However, they should state why these five models were specifically selected from a wide range of other possible numerous deep learning models (e.g. Recurrent Neural Network (RNN), Gated Recurrent Unit (GRU), Temporal Convolutional Network (TCN), Transformer, Seq2Seq (Sequence-to-Sequence) Model, DeepAR, TFT (Temporal Fusion Transformer), WaveNet, ConvLSTM, Multi-layer Perceptron (MLP) for Time Series, Deep State Space Models, DeepTCN (Deep Temporal Convolutional Networks), Variational Autoencoders (VAE) for Time Series, etc.). I am not suggesting including these models in your work but state the reason for the five models selected for this work. Are your selected five model superior to others? Does better at time series forecasting?

Section 3.2.3

• It is not clear what the authors mean by “years before 2017, 2018, 2019, 2020, and 2021”. Are they referring to years from 1999 to 2016? Or 2012, 2013, 2014, 2015, 2016?

Section 3.3

• Since the tables 6-9 contain a lot of data, the authors should provide more guidance to the readers on how to interpret the results in these tables. Either use short footnotes under the table to describe the data or embed some explanation in the text to aid readers’ understanding.

o For example, what does 0.99-1.0, 0.9-1.00,…etc., mean?

o Is the reader supposed to compare each column against “single” column?

o Do the % mean prediction error or accuracy? Are high % values good or bad?

o Any explanation for why the Average RSME (and Average MAPE) value decrease from left to right in Table 8 but increase from left to right in table 7. On the other hand, the values fluctuate in table 6 and 9. Please give adequate explanation

o “As shown in Tables 6 to 9, models with added features exhibit better prediction accuracy than the "Single" group”. By this statement, are you comparing the average values of each column (0.99-1.0, 0.9-1.00,…etc ) vs average of “single” column? This needs to be clarified, because some of the column values within each row are larger/smaller than the “single value”

o Overall, this section needs re-working

Section 3.4

• Line 8/9, generation of oil is repeated twice

• Correct “Important” in Table 12 and 14 headers to “Import”

• Ensure the % accuracies in lines 7/8 match the accuracies of each of the four indicators in Tables 11 to 14. See comment on conclusions section below for similar mistake.

• Grammar: “In conclusion, we conclude” is redundant. Perhaps you can update to “In conclusion, the identified significant feature…”

• What does “differencing” mean in the explanation of Arima (Table 10)

Section 3.5

• Is the “Experimental Details” in the first line of this section a subheading? Clarify or find a better way to integrate it into the rest of the sentences in that paragraph

• Only 4 ensemble models are listed in line 2 (instead of 5). The authors must have omitted XBOOST. Check and revise accordingly

• Check 101 neurons.we set the batch. There should be a space between “neurons” and “we”.

• The authors should specify in the table (refer readers to the table row/column) the one instance where the ensemble model performs worse than the LSTM model

• Grammar: “In the 100 comparison cases” is somewhat ambiguous. One possible adjustment will be “For each of the four indicators, 25 cases were run, giving a total of 100 cases, as show in Fig 15”

•

Section 3.6

• Grammar: Change beside to besides.

• Consider using the same vertical scale for Figure 5, 7 and any other figures which have similar data range in the vertical axis.

Section 4 (Conclusion)

• There is a mismatch % accuracy. For example, in table 11, the generation of oil accuracy improves by 2.2% not 2.12% stated in the conclusion. The authors should reconcile these % accuracies in the conclusion section to reflect the correct values in table 11

• “Generation of oil” is repeated twice in line 3 and 4 of your conclusion. The same error also in bullet point 2 of conclusion. This occurs in other places within the document. The authors should carefully go through the document to correct all of them.

• Grammar: We have evaluated our approach’s forecasting performance for… to “we evaluated the performance of our approach using four indictors…”

In General

• It is not clear what the authors mean by “ablation”, for example, “oil ablation”. Another word/synonym should be used instead to convey their meaning

• Consider using “Oil production” in place of “Oil generation”.

• Words like “competitors” is not appropriate for an academic journal publication. While it is good to benchmark your work against other authors/researchers, referring to them as competitors is not appropriate, as this is not a business endeavor. The authors should find better synonyms for “competitor” in the journal

• While it is good to compare the ensemble model against others using indices like loss/win, “prediction errors”, F-rank, etc., the authors should include an actual time series forecast. For example, take oil import indicator vs time, plot the real data vs ensemble model forecast, to show how their ensemble model performs vs real data.

• The use of prediction error and accuracy is somewhat ambiguous in the article. The authors should adopt one and stick to it

6. PLOS authors have the option to publish the peer review history of their article (what does this mean?). If published, this will include your full peer review and any attached files.

Reviewer #1: **Yes: **Brooke Rogachuk

Reviewer #2: No

---

## [Author Response · Author response to Decision Letter 0]

14 Sep 2024

Associate Editor

Comments: Ensure that words such as Novel, new study, first time are removed from the manuscript. Also clarify the meaning of hybrid deep learning methods, all figures should be clear. Authors should also explain the difference between hybrid and stacked ML models. Review for grammatical and spelling errors.

Response: The authors highly appreciate the AE’s recommendation and comments. We have carefully revised the manuscript according to the Reviewers’ comments one by one. Words such as Novel, new study, first time have been removed from the manuscript. The meaning of hybrid deep learning methods has been clarified to make our proposed model clear to be un-derstood. All figures have been replaced by clear versions. We have proofread the whole manuscript with care to address its grammatical and spelling issues to improve its readability. Please review the revised manuscript. Thank you very much for handling this paper.

Reviewer #1

RQ1. The research presented in this paper is novel and yields significant results, but it re-quires substantial reshaping to be clear and impactful. In the abstract, the repeated mention of "generation of oil" should be corrected to "generation of coal" to accurately reflect the four targeted indicators. Additionally, more detail on what differentiates the five deep learning models would help readers understand the ensemble's diversity and benefits. Briefly mention-ing the types of datasets used, such as the time span and data sources, would also provide clearer context for the experiments and enhance the abstract's informativeness.

Response: The authors highly appreciate the Reviewer for pointing these issues out. Follow-ing your comments, we have rewritten the corresponding contents in the Abstract. Please refer to the Abstract of the revised manuscript.

RQ2. The introduction could benefit from a revision of the first sentence for greater clarity, setting a better stage for the research. The data presentation throughout the paper has several issues that need addressing. The constant use of bullet points is distracting; a more narrative style would improve readability and flow. 

Response: The authors highly appreciate the Reviewer for these excellent comments. We have carefully revised the Introduction and the Data Presentation and have deleted some bullet points to adopt the narrative style to improve the readability. Please review the revised manuscript.

RQ3. Table 1 is too dense with text, focuses on only one point, and thus is less effective. I do not understnad why that data presented is even in a table. Simplifying and summarizing the information will make it more reader-friendly.

Response: The authors highly appreciate the Reviewer for this excellent comment. We have deleted the original Table 1. The introduction of the collected data has been revised. Please refer to the section 3.1 of the revised manuscript. 

RQ4. Table 2's placement in the paper is confusing and should be reconsidered for better logical flow. Moreover, tables 2 through 5 should be grouped together if they are related, and adding lines to these tables would improve readability. Each table should be accompanied by a general statement explaining its content and significance.

Response: The authors highly appreciate the Reviewer for this excellent comment. We have reorganized these tables as well as made more explanations. Please refer to Tables 1-4 of the revised manuscript.

RQ5. Figures 2 and 5 are blurry and hard to read, necessitating clearer visuals to effective-ly communicate the data. There is also insufficient explanation for each figure and table; more detailed descriptions rather than the current one-sentence explanations are needed. 

Response: The authors highly appreciate the Reviewer for pointing these issues out. All fig-ures have been replaced by clear versions. Besides, we have provided more explanations for all the figures and tables. Please review the revised manuscript.

RQ6. In Section 3.2.1, the bullet points are unnecessary and should be revised into a more cohesive narrative. Section 3.4 needs to be reworked entirely; the results discussed in the tables should be expanded upon, integrating the discussion into the overall implementation and im-plications of the work.

Response: The authors highly appreciate the Reviewer for these excellent comments. Fol-lowing your comments, we have rewritten the sections 3.2.1 and 3.4. Please refer to sections 3.2.1 and 3.4 of the revised manuscript.

RQ7. The summary requires several improvements. The repeated mention of "generation of oil" should be corrected to accurately reflect the four indicators: import of oil, generation of oil, import of coal, and generation of coal. Additionally, providing a brief explanation or example of what feature engineering involves would be beneficial, as not all readers may be familiar with the term. More detail on the specific deep learning models used in the ensemble would also give a fuller picture of the methodology and its advantages.

Response: The authors highly appreciate the Reviewer for these excellent suggestions. Fol-lowing your comments, we have rewritten the entire section 4. Conclusion. Please refer to the section 4 of the revised manuscript.

RQ8. By addressing these points, the paper can better communicate its novel approach and significant findings, making it a clearer and more impactful contribution to the field of energy consumption forecasting. I cannot emphasize enough the importance of having more explana-tion of these tables and figures.

Response: The authors highly appreciate the Reviewer for your insightful and informative comments, which are very helpful for significantly improving our manuscript. We have carefully revised this manuscript according to your comments one by one. We would appreciate if you could review this paper again.

Reviewer 2

The authors have clearly demonstrated that the use of ensemble models is effective in addressing the limitations of single models in time series forecasting of heterogeneous indicators, including the superiority of using multi-feature data over univariate data in enhancing prediction accuracy. I recommend the acceptance of the paper for publication, subject to some revisions below. Overall, the paper needs some improvement/polishing of the grammar, especially the introduction. The authors should revise the grammar of the manuscript for better readability (I have highlighted JUST A FEW (as “Grammar” below – there is more.)

Response: The authors highly appreciate the Reviewer’s encouraging comments. The manu-script has been carefully revised with care according to your comments one by one. Detailed responses are provided below. 

RQ1. Introduction. The second paragraph in this section should be re-written for better flow of thoughts. I find it hard to connect all the different statistical analysis methods and how one relates and/or build on the other. Each sentence in this section sounds disjointed from the pre-ceding and/or proceeding sentence. Consider using words like “nevertheless”, “also”, “how-ever”, “moreover”, “in addition” to connect the various statistical methods, and how they re-late to each other.

Response: The authors highly appreciate the Reviewer for this excellent comment. Following your comment, we have revised the second paragraph of introduction. Please review the cor-respond contents in the revised manuscript.

RQ2. Introduction. Paragraph 6. Rather than simply cite the other previous authors who have implemented ensemble modelling to energy related forecasting, the authors should elabo-rate more on these previous works (approach, outcomes, shortcoming). This will help build a case for the uniqueness of this article vs previous studies. For example, what makes your ap-proach unique compared to other authors who have implemented ensemble modelling for en-ergy related forecasting?

Response: The authors highly appreciate the Reviewer for this excellent comment. Following your comment, we have revised paragraph 6 of the introduction to highlight the innovation of this study. Please review the corresponding contents in the revised manuscript.

RQ3. Introduction. Grammar: “Their consumption forecasting is crucial for China because it can not only provide a clear understanding of the future energy situation, but also help gov-ernments optimize and adjust its energy strategies to ensure energy security”, can be re-written as “Consumption forecasting is crucial in China as it not only provide a clear understanding of the future energy landscape but also helps the government to optimize and adjust strategies, thereby ensuring energy security”. Grammar: Ensemble model combines a team of diverse models…Ensemble model combines a collection of diverse models.

Response: The authors highly appreciate the Reviewer for pointing these issues out. We have fixed these grammar errors. Please review the revised manuscript.

RQ4. Section 2.3. The authors have done an excellent job in describing the five forecasting models. However, they should state why these five models were specifically selected from a wide range of other possible numerous deep learning models (e.g. Recurrent Neural Network (RNN), Gated Recurrent Unit (GRU), Temporal Convolutional Network (TCN), Transformer, Seq2Seq (Sequence-to-Sequence) Model, DeepAR, TFT (Temporal Fusion Transformer), WaveNet, ConvLSTM, Multi-layer Perceptron (MLP) for Time Series, Deep State Space Mod-els, DeepTCN (Deep Temporal Convolutional Networks), Variational Autoencoders (VAE) for Time Series, etc.). I am not suggesting including these models in your work but state the reason for the five models selected for this work. Are your selected five model superior to others? Does better at time series forecasting?

Response: The authors highly appreciate the Reviewer for these excellent questions. A single deep learning model can not comprehensively capture all the inherent characteristics of dif-ferent indicators. Ensemble learning is an effective way to address this issue by combining several deep learning models. However, ensemble learning requires that the base deep learning models have the characteristics of diversity and accuracy. Diversity: the individual models need to make different kinds of predictions, which ensures that when one model makes a wrong prediction, another can potentially correct it. Accuracy: the individual models should still be reasonably accurate. If each model is weak (too inaccurate), the ensemble won’t per-form well, even with diversity. Following this principle, we employed the five typical deep learning models of LSTM [21], XGBOOST [30], TCN [31], CNN [32], and TRMF [33] for ensembling. First, they have different learning structures and forecasting mechanisms to make them diverse. Second, they have been demonstrated to be accurate and effective in forecast-ing energy-related time-series. Hence, the ensembled model enjoys all the merits of the five deep learning models, making it able to comprehensively capture all the characteristics of dif-ferent indicators to achieve accurate forecasting. The above explanations have also been added in section 2.3 of the revised manuscript. 

RQ5. Section 3.2.3. It is not clear what the authors mean by “years before 2017, 2018, 2019, 2020, and 2021”. Are they referring to years from 1999 to 2016? Or 2012, 2013, 2014, 2015, 2016?

Response: The authors highly appreciate the Reviewer for pointing these issues out. In the experiments, the training sets were constructed by using the data from 1999 to 2016, and the remaining five years data (i.e., 2017, 2018, 2019, 2020, and 2021) were set as testing sets. We have revised this issue in the section 3.2.2 of the revised manuscript.

RQ6. Section 3.3. Since the tables 6-9 contain a lot of data, the authors should provide more guidance to the readers on how to interpret the results in these tables. Either use short footnotes under the table to describe the data or embed some explanation in the text to aid readers’ understanding. 

o For example, what does 0.99-1.0, 0.9-1.00,…etc., mean?

o Is the reader supposed to compare each column against “single” column?

o Do the % mean prediction error or accuracy? Are high % values good or bad?

o Any explanation for why the Average RSME (and Average MAPE) value decrease from left to right in Table 8 but increase from left to right in table 7. On the other hand, the values fluctuate in table 6 and 9. Please give adequate explanation.

o “As shown in Tables 6 to 9, models with added features exhibit better prediction accuracy than the "Single" group”. By this statement, are you comparing the average values of each column (0.99-1.0, 0.9-1.00,…etc ) vs average of “single” column? This needs to be clarified, because some of the column values within each row are larger/smaller than the “single value”

o Overall, this section needs re-working

Response: The authors highly appreciate the Reviewer for these excellent comments. Re-garding the raised issues, we make the following clarifications:

(1) 0.99-1.0 and 0.9-1.00 denotes that the features with correlation coefficients of 0.90-1.00 were grouped or used to model.

(2) The column of ‘Single feature’ denotes that only the targeted indicator (without features) was used to model for forecasting.

(3) % denotes the MAPE. Lower MAPE denotes higher forecasting accuracy. 

(4) For different indicators, the optimal features are different. Hence, the highest forecasting accuracy of different indicators corresponds to the different columns that have different feature correlation coefficients. The highest forecasting accuracy of each indicator has been highlighted in tables 5-8. 

(5) The column of ‘Single feature’ denotes that only the targeted indicator (without features) was used to model for forecasting. 

Following your comments, we have carefully rewritten this section. More explanations and discussions have been provided. Please review the section 3.3 in the revised manuscript.

RQ7. Section 3.4. Line 8/9, generation of oil is repeated twice. Correct “Important” in Ta-ble 12 and 14 headers to “Import”. Ensure the % accuracies in lines 7/8 match the accuracies of each of the four indicators in Tables 11 to 14. See comment on conclusions section below for similar mistake. Grammar: “In conclusion, we conclude” is redundant. Perhaps you can update to “In conclusion, the identified significant feature…”. What does “differencing” mean in the explanation of Arima (Table 10).

Response: The authors highly appreciate the Reviewer for pointing these issues out. We have carefully revised these issues in the revised manuscript.

RQ8. Section 3.5. Is the “Experimental Details” in the first line of this section a subhead-ing? Clarify or find a better way to integrate it into the rest of the sentences in that paragraph. Only 4 ensemble models are listed in line 2 (instead of 5). The authors must have omitted XBOOST. Check and revise accordingly. Check 101 neurons. we set the batch. There should be a space between “neurons” and “we”. The authors should specify in the table (refer readers to the table row/column) the one instance where the ensemble model performs worse than the LSTM model. Grammar: “In the 100 comparison cases” is somewhat ambiguous. One possible adjustment will be “For each of the four indicators, 25 cases were run, giving a total of 100 cases, as show in Fig 15”.

Response: The authors highly appreciate the Reviewer for pointing these issues out. We have carefully revised these issues in the revised manuscript.

RQ9. Section 3.6. Grammar: Change beside to besides. Consider using the same vertical scale for Figure 5, 7 and any other figures which have similar data range in the vertical axis.

Response: The authors highly appreciate the Reviewer for pointing these issues out. We have carefully revised these issues. In the revised manuscript, Fig 6 has a small range difference in vertical scale because their highest MAPEs are different.

RQ10. Section 4 (Conclusion). There is a mismatch % accuracy. For example, in table 11, the generation of oil accuracy improves by 2.2% not 2.12% stated in the conclusion. The au-thors should reconcile these % accuracies in the conclusion section to reflect the correct values 

---

## [Decision Letter · Decision Letter 1]

11 Oct 2024

PONE-D-24-27898R1Energy Consumption Forecasting for Oil and Coal in China Based on Hybrid Deep LearningPLOS ONE

Dear Dr. Wu,

Thank you for submitting your manuscript to PLOS ONE. After careful consideration, we feel that it has merit but does not fully meet PLOS ONE’s publication criteria as it currently stands. Therefore, we invite you to submit a revised version of the manuscript that addresses the points raised during the review process.

The Y axis of figure 6  (previously figure 7 in the original manuscript) as contained in the reviewer’s feedback comment was still not corrected. Should be corrected before publication!

Inconsistent citation style. 

We look forward to receiving your revised manuscript.

Kind regards,

Jude Okolie, Ph.D.

Academic Editor

PLOS ONE

Journal Requirements:

Reviewers' comments:

Reviewer's Responses to Questions

**Comments to the Author**

1. If the authors have adequately addressed your comments raised in a previous round of review and you feel that this manuscript is now acceptable for publication, you may indicate that here to bypass the “Comments to the Author” section, enter your conflict of interest statement in the “Confidential to Editor” section, and submit your "Accept" recommendation.

Reviewer #1: All comments have been addressed

Reviewer #2: (No Response)

2. Is the manuscript technically sound, and do the data support the conclusions?

Reviewer #1: Yes

Reviewer #2: (No Response)

3. Has the statistical analysis been performed appropriately and rigorously? 

Reviewer #1: Yes

Reviewer #2: (No Response)

4. Have the authors made all data underlying the findings in their manuscript fully available?

Reviewer #1: Yes

Reviewer #2: (No Response)

5. Is the manuscript presented in an intelligible fashion and written in standard English?

Reviewer #1: Yes

Reviewer #2: (No Response)

6. Review Comments to the Author

Reviewer #1: The authors met all of my comments! I believe this is now a novel and interesting study and the paper reflects that!

Reviewer #2: Overall comment: The authors have implemented majority of the comments/concerns raised in the first manuscript. This article should be accepted for final publication, subject to the following minor revisions.

Table 14. The case of our approach loses the comparison is row of ‘2020’ in “Production of coal, not “Import of oil”

The authors have didn’t change the y-axis range in Figure 6 (previously figure 7 in the original manuscript) as contained in the reviewer’s feedback comment. Should be corrected before publication!

Mix of citation style in Introduction paragraph 2. E.g Tong et al(date?). and Rao et al(date?) is used together with numbered citations (e.g [1], [2], [3],…etc). If this is acceptable to the journal, fine. Otherwise, the authors should correct/unify these before final publication

7. PLOS authors have the option to publish the peer review history of their article (what does this mean?). If published, this will include your full peer review and any attached files.

Reviewer #1: **Yes: **Brooke E. Rogachuk

Reviewer #2: No

---

## [Author Response · Author response to Decision Letter 1]

13 Oct 2024

Associate Editor

Comments: The Y axis of figure 6 (previously figure 7 in the original manuscript) as contained in the reviewer’s feedback comment was still not corrected. Should be corrected before publication!

Response: The authors highly appreciate the AE’s recommendation and comments. We have revised the Y axis of figure 6. Please note that all the modifications we’ve made are colored blue in the revised manuscript. Thank you very much for handling this paper.

Reviewer #1

RQ1. The authors met all of my comments! I believe this is now a novel and interesting study and the paper reflects that!

Response: The authors highly appreciate the Reviewer for your recommendation.

Reviewer 2

The authors have implemented majority of the comments/concerns raised in the first manu-script. This article should be accepted for final publication, subject to the following minor revi-sions. 

Response: The authors highly appreciate the Reviewer’s encouraging comments. The manu-script has been carefully revised with care according to your raised minor revisions. Detailed responses are provided below. 

RQ1. Table 14. The case of our approach loses the comparison is row of ‘2020’ in “Produc-tion of coal, not “Import of oil” 

Response: The authors highly appreciate the Reviewer for pointing this issue out. We have fixed this issue. Please review the revised manuscript.

RQ2. The authors have didn’t change the y-axis range in Figure 6 (previously figure 7 in the original manuscript) as contained in the reviewer’s feedback comment. Should be correct-ed before publication! 

Response: The authors highly appreciate the Reviewer for pointing this issue out. We have changed the y-axis range in Figure. Please review the revised manuscript.

RQ3. Mix of citation style in Introduction paragraph 2. E.g Tong et al(date?). and Rao et al(date?) is used together with numbered citations (e.g [1], [2], [3],…etc). If this is ac-ceptable to the journal, fine. Otherwise, the authors should correct/unify these before final publication

Response: The authors highly appreciate the Reviewer for pointing this issue out. We have revised this issue in Introduction. Please review the revised manuscript.

---

## [Decision Letter · Decision Letter 2]

1 Nov 2024

Energy Consumption Forecasting for Oil and Coal in China Based on Hybrid Deep Learning

PONE-D-24-27898R2

Dear Dr. Wu,

We’re pleased to inform you that your manuscript has been judged scientifically suitable for publication and will be formally accepted for publication once it meets all outstanding technical requirements.

Kind regards,

Jude Okolie, Ph.D.

Academic Editor

PLOS ONE

Additional Editor Comments (optional):

Reviewers' comments:

Reviewer's Responses to Questions

**Comments to the Author**

1. If the authors have adequately addressed your comments raised in a previous round of review and you feel that this manuscript is now acceptable for publication, you may indicate that here to bypass the “Comments to the Author” section, enter your conflict of interest statement in the “Confidential to Editor” section, and submit your "Accept" recommendation.

Reviewer #1: All comments have been addressed

Reviewer #2: (No Response)

2. Is the manuscript technically sound, and do the data support the conclusions?

Reviewer #1: Yes

Reviewer #2: (No Response)

3. Has the statistical analysis been performed appropriately and rigorously? 

Reviewer #1: Yes

Reviewer #2: (No Response)

4. Have the authors made all data underlying the findings in their manuscript fully available?

Reviewer #1: Yes

Reviewer #2: (No Response)

5. Is the manuscript presented in an intelligible fashion and written in standard English?

Reviewer #1: Yes

Reviewer #2: (No Response)

6. Review Comments to the Author

Reviewer #1: Figure 6 has been fixed, and everything looks fantastic now! This paper is not only well-structured but also presents some truly novel insights. Great work!

Reviewer #2: (No Response)

7. PLOS authors have the option to publish the peer review history of their article (what does this mean?). If published, this will include your full peer review and any attached files.

Reviewer #1: **Yes: **Brooke Rogachuk

Reviewer #2: No

---

## [Editor Report · Acceptance letter]

7 Nov 2024

PONE-D-24-27898R2 

PLOS ONE

Dear Dr. Wu, 

I'm pleased to inform you that your manuscript has been deemed suitable for publication in PLOS ONE. Congratulations! Your manuscript is now being handed over to our production team.

Kind regards, 

on behalf of

Dr. Jude Okolie 

Academic Editor

PLOS ONE